# Modeling and Simulation of Processes in a Factory of the Future

**Patrik Grznár [1,*][ID], Milan Gregor [1], Martin Krajčovič [1], Štefan Mozol [1], Marek Schickerle [1], Vladimír Vavrík [1], Lukáš Ďurica [2], Martin Marschall [2] and Tomáš Bielik [2]**

1. Department of Industrial Engineering, Faculty of Mechanical Engineering, University of Žilina, Univerzitná 8215/1, 010 26 Žilina, Slovakia; milan.gregor@fstroj.uniza.sk (M.G.); martin.krajcovic@fstroj.uniza.sk (M.K.); stefan.mozol@fstroj.uniza.sk (Š.M.); schickerle@stud.uniza.sk (M.S.); vladimir.vavrik@fstroj.uniza.sk (V.V.)

2. Institute of Competitiveness and Innovations, University of Žilina, Univerzitná 8215/1, 010 26 Žilina, Slovakia; lukas.durica@fstroj.uniza.sk (L.Ď.); martin.marschall@fstroj.uniza.sk (M.M.); tomas.bielik@fstroj.uniza.sk (T.B.)

\* Correspondence: patrik.grznar@fstroj.uniza.sk; Tel.: +421-41-513 2733



**Featured Application: Application of article is mainly in the area of future manufacturing systems where the control system will use simulation for predicting future state and base on information carry out actions.**

**Abstract:** Current trends in manufacturing, which are based on customisation and gradually customised production, are becoming the main initiator for the development of new manufacturing approaches. New manufacturing approaches are counted as the application of new behavioural management patterns that calculate the retained competencies of decision-making by the individual members of the system agent; the production becomes decentralised. The interaction of the members of such a system creates emergent behaviour, where the result cannot be accurately determined by ordinary methods and simulation must be applied. Modelling and simulation will, therefore, be an integral part of the planning and control of the processes of factories of the future. The purpose of the article is to describe the use of modelling and simulation processes in factories of the future. The first part of the article describes new manufacturing concepts that will be used in factories of the future, with a description of modelling and simulation routing in the frame of Industry 4.0. The next section describes how simulation is used for the control of manufacturing processes in factories of the future. The included subsection describes the implementation of this suggested pattern in the laboratory of ZIMS (Zilina Intelligent Manufacturing System), with an example of a metamodeling application and the results obtained.

**Keywords:** advanced industrial engineering; modelling and simulation; factory of the future; smart factory; manufacturing systems; production planning optimisation; decision support

## 1. Introduction

Future manufacturing systems will differ significantly from those of today. The changes will not only result in the pressure of customers on the variant of new products but also revolutionary changes in the impact of technological innovation. The most significant factor that affects the existing manufacturing environment is the customer. The factory must be able to produce the required product in the shortest possible time and at a reasonable cost. Future manufacturing will provide products that will be tailored to the requirements of a particular customer, highly sophisticated, complex, and capable of offering new functionality; therefore, it will require an entirely new manufacturing environment.

The customisation and personalisation of products are a complex problem that researchers are trying to tackle today. On the one hand, researchers have used the appropriate construction of new products, also known as modular, reconfigurable products. On the other hand, they have also tried to increase the flexibility of the manufacturing system, which we now refer to as reconfigurable manufacturing. However, future manufacturing systems will use completely new principles in their operation. Researchers have sought to develop and exploit new methods and approaches to product design and production due to the growing complexity of both products and manufacturing systems [1].

Industry 4.0 is a digital revolution being witnessed in the present generation, whereby the aim is to digitise the entire manufacturing process with minimal human or manual intervention [2]. We are in a time where every major breakthrough in technology changes the face of manufacturing industries. At present, we are in the era of Industry 4.0, which is hailed as the age of cyber-physical systems (CPSs) that has taken manufacturing and associated industry processes to an unforeseen level with flexible production, including manufacturing, supply chain, delivery, and maintenance [3]. The development of the Industry 4.0 concept was needed to develop new competitive business models. These business models need to be based on cooperation and better use of the available resources [4]. Industry 4.0 is based on digitalisation and application of exponential technologies. Digitisation and application of exponential technologies are directly linked to CPSs. CPSs presaturate physical devices with built-in tools for digital data collection, processing, and distribution, and, through the internet, are connected to each other online. CPSs form the basis for technology such as the Internet of Things and, in combination with the Internet of Services, form the base for Industry 4.0.

New factories, or their manufacturing systems, will have unique features that enable them to respond quickly and efficiently to frequently changing customer demands. These manufacturing systems will be designed as modular, reconfigurable, and intelligent holonic systems capable of rapidly changing their functions and capacities based on the auto diagnostic. The dynamism of complex manufacturing systems will no longer be possible to study using today's modelling and simulation techniques. The future dynamic manufacturing environment will require robust modelling and simulation tools that will be able to simulate complex phenomena and processes. New simulation systems must function as part of complex control systems, working in real-time and must be used to support decision-making and the creation of new knowledge. In this case, real-time work is seen as a rapid response to emerging events and time deterministic calculation of the trajectories of the development of future manufacturing system conditions [5].

Simulation has become the most essential tool for dynamic analysis of complex systems in recent decades. A high level of development has mainly seen a discreet simulation using the principles of the event orientation. The latest simulation tools have thus simplified the process of creating simulation models that today are being waived from the use of more straightforward analytical methods [6]. Today, artificial intelligence or virtual reality is the usual supportive technique used in simulations. The importance of simulation grows mainly with the increasing complexity of systems. They are mainly used where an erroneous decision can mean inefficient investment, long-term economic losses, and a weakening of its competitiveness.

In the growth of systems complexity and deployment of smart devices that decide on actions in factories of the future, it is, therefore, necessary to determine the outcome of the actions for management needs in a high emergence of processes. [7]. The requirement of frequent changes to the production base requires the rapid commissioning systems of automated manufacturing systems (Ramp Up), which will require new simulation tools. In the case of control, emulating technologies that are tied to the simulation may be used. One of the advantages of the emulation environment is that it can monitor the technical system (such as production, assembly, logistics) in real-time to evaluate the data collected and to update the model in question on a real-system basis and to carry out experiments on the simulation model simultaneously. In Industry 4.0, the introduction of the digital twinning of objects and processes is equally important [8].

The orientation of research into new manufacturing approaches is directed towards the area of intelligent manufacturing systems, using reconfigurable manufacturing systems, adaptive logistics, and the concept of competence islands. New simulation systems must also be adapted to this new requirement. They must possess the ability to simulate agent systems and model large networks. Modelling and simulation will, therefore, be an integral part of the planning and control of the processes of factories of the future.

Research on the principles of modelling and simulation and the development of factories of the future have been the long-term areas of research at the Department of Industrial Engineering, University of Žilina. The issue addressed is consistent with the strategy of Industry 4.0. Just by defining the characteristics of the systems used in factories of the future and their properties can be evaluated, as such systems can be modelled and simulated. The article, in its periphery, deals with the description of the manufacturing concepts that are potentially highly applicable for use in factories of the future, and the core descriptions of the use of modelling and simulation, mainly metamodeling, in the processes control of factories of the future. An example of using this approach is described in the processes control of laboratory ZIMS (Zilina Intelligent Manufacturing System).

## 2. Materials and Methods

### 2.1. Changing of Business Priorities of Factories

The changing demands of customers and emerging progressive technologies are revolutionising not only the existing manufacturing environment but, at the same time, bringing about a change in the main paradigm of business. Business priorities are dynamically changing. The technical level of production factories and systems for the planning and control of the activities of these factories fundamentally affect the productivity and efficiency of each factory and hence its competitiveness.

In the past, when world markets were not saturated, it was a priority for businesses to achieve high production capacity utilisation. Capacities accounted for capital invested, and businesses tried to assess it. The aim was to produce simple products in high production volumes (mass), which guaranteed the benefits from the economy of the quantity. Businesses were looking for low-wage territories, which triggered a mass shift to countries with cheap labour, known as offshoring. After the market was saturated, the customer's requirements and preferences gradually changed, resulting in increasing product variants and the combined growth of production complexity. Gradually, the priority of high capacity utilisation and low wages has been replaced by the priority of high flexibility and management of complexity in the production.

The flexibility of production is a prerequisite for the production of a wide range of different products. Thus, the priority of the high capacity utilisation was replaced by the requirement of the high flexibility of production and the ability to cope with the complexity of such production. Only this approach guarantees sustained productivity growth. The relationship between flexibility and productivity is shown in Figure 1.

**Flexibility versus Productivity Relationship**

**Figure 1.** Flexibility versus productivity relationship [9].

Since the late 20th century, the latest information and communication technologies have begun to be mass-deployed in manufacturing companies, resulting in cheaper means of automation and industrial robots. The cost of automated production was lower than labour costs, and it brought a new phenomenon, now known as reshoring. Businesses began to move their production capacity from countries with cheap labour to their parent countries [10]. A good example is large corporations from the US (Intel, General Motors, General Electric).

The company is, therefore, transformed from production to services. New ways and forms of value creation are created. This brings significant changes to today's factories. The classic factories and their manufacturing systems are gradually transformed, as they end up with classic products we have perceived for centuries. Customers today do not need to possess physical product models when they look for a service that fills their requirements. The young generation no longer needs things to possess. They want to share them, thus stimulating the emergence of a so-called sharing economy. The importance of leasing has grown. Competition is shifting from products to business models. Today, businesses are already competing with their business models [9]. The price of the product is still the determining criterion by which most customers choose from the offerings. If several manufacturers offer similar products, the selling price is the only criterion in their decision-making.

The objective is the simple integrability of product variants in customer-oriented manufacturing systems and, consequently, the efficient implementation of mass production, so that the production of an affordable product is ensured through the economies of scale. It is known that the reduction of unit costs is due to the increasing volume of products produced. There are usually several global producers in the markets that offer similar products with a similar level of quality at relatively low prices. Small firms can compete in such an environment only by finding a more efficient production method, similar to that used by large producers. If the manufacturer wants to sell more expensive products, it must bring new value to the market, e.g., new products with different characteristics that customers will appreciate and buy. The new customer requirements are thus linked to the growth of product variants. The highest form of satisfaction of customer requirements is the personalisation of demand. This means that each product is tailor-made to the customer. The strategy of mass customisation may be appropriate for the economies of scope, and its effective implementation is not possible without advanced manufacturing systems capable of responding rapidly to changes [11]. These changes affect many of the long-established patterns of behaviour, known as paradigms.

### 2.2. Paradigm Changes

We live in a time when paradigm changes are underway, and their main drivers are new, emerging technologies. Advanced technologies affect the life of the whole society. Their most significant impact is reflected in the production sphere. Robotics is one of the areas that have, for several decades, undergone a technological revolution. Industrial robots have already become commonplace in production. The development and deployment of mobile robots in production and logistics are also on a similar path. Over the past five years, research labs have found their way into manufacturing workshops and cooperative robots (cobot). These represent an intervertebral stage in the transformation of manufacturing systems by integrating the activity of man and robot cooperatively.

At this stage, the man remains part of the production processes. Another development step is humanoid robots, which are gradually becoming a priority in research and quickly penetrating manufacturing practice and services. One of the main priorities for research is collective robotics, namely, the control and coordination of the target behaviour of a group of heterogeneous robots. Manufacturing systems and overall production are in the permanent transformation phase.

The current driving forces of the development of future enterprises by [9] can be classified into two independent groups, as shown in Figure 2.

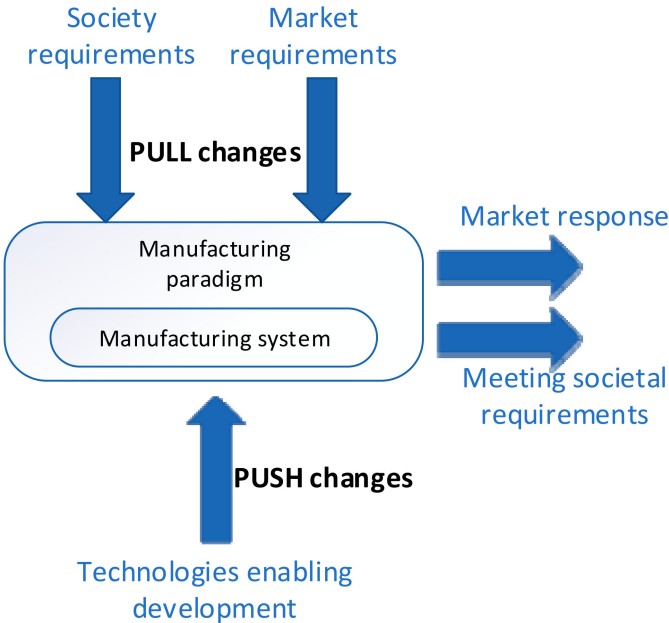

**Figure 2.** Illustration of PULL and PUSH changes [12].

Emerging Technologies (EmTe)—these are technologies that are often from the outset overlooked and, later, can transform entire industries or services sectors. These drivers are also called PUSH-changes because the changes are due to a technological push.

Market changes (changes in customer requirements)—these changes can often be tracked in long-term trends. Customers demand products (services) that better and more precisely satisfy their requirements, which manufacturers implement through the customisation and personalisation of products. As it is the customers who make this move, we call these changes "PULL changes".

In conventional manufacturing systems as a transfer line, lean manufacturing and flexible manufacturing systems may be advantageous to mass production.

In the mass production of standard products, the primary source of the competitive advantage is transfer lines for the production of one product type using fixed jigs and tools. The aim is to produce one product type in large production volumes and the required production quality, which means low production costs.

The concept of lean manufacturing was created at Toyota as an extension of the functions of mass manufacturing. It is also known as the Toyota Production System, and its primary purpose was to reduce the production lead time, linked with quality growth, cost reduction and loss-making and wastage.

Flexible manufacturing systems (FMSs) have been designed as solutions to ensure sufficient production of the family of similar products. FMSs use NC-controlled machines, automatic handling and transport systems and integrated production management. The deployment of information technologies with the possibility of rapid reprogramming has allowed the product range to be easily changed.

Conventional manufacturing systems used to control the production of classic PUSH approaches and, later, PULL-control. This manufacturing system was, at the time of origin, sufficient. However, now, some areas of markets in which conventional manufacturing systems are used have started to require a variety of products, with little time of product placement in the market. Hence, this growing demand of customers has created a complexity of production that has prompted the increasing experimentation of development of reconfigurable production lines or so-called competence islands.

The paradigm change that is currently occurring is characterised by the use of agents and the principles of multi-agent control in manufacturing. In practice, this means that the classic control systems will be progressively replaced by multi-agent control. Multi-agent control brings the manufacturing of emergence, which means that the characteristics of manufacturing systems are also changing as they become emergent.

The traditional manufacturing systems were complicated. The multi-agent control application in manufacturing represents a transformation of complicated systems into complex systems. For complex systems, the complexity of interrelationships between the various elements of the system is already so significant that it often tends to be very demanding, if not impossible, to use mathematical modelling for their studies. The dynamic behaviour of such systems can only be studied using the theory of complexity. Future manufacturing systems will operate as adaptive, dynamic manufacturing networks. New simulation systems must also be adapted to this new requirement. They must possess the ability to simulate agent systems and modelling large networks. Modelling and simulation will, therefore, be an integral part of the planning and control of the processes of factories of the future. In manufacturing, in addition to real objects, there will also be their virtual representatives, which we now refer to as digital twins. Such a dual representation of production is also known as virtual manufacturing. For the visualisation of future manufacturing systems, we can see a similarity to living organisms. Holonic production with multi-agent control will resemble more the emergence of the functioning of living organisms rather than a mechanical automaton.

A new trend in manufacturing systems development is reconfigurable manufacturing systems. Nature teaches us that when changing the environment, the living organism strives to adapt to changed conditions. It uses the change of internal structures and the number of elements and their composition. At the molecular level, it "stretches and recycles" unnecessary structures and reconfigures them into new, necessary structures. Recycling is a process of decomposition. Reconfiguration then represents the new use of existing structures [13].

Most of the activities in the industry of the future will be performed by intelligent robots. In order for the robots to be able to carry out their tasks, often in an unfamiliar environment, they have to possess autonomous capabilities, hence, the ability to adapt to their surroundings and the changing conditions of the surrounding area, collect and evaluate information about their internal state and environment (perception), predict future situations, make the necessary decisions and, of course, learn from the situations. Such tasks can now be tackled by the individual, advanced robotic systems.

The growing interest in mobile robotics applications has not only made changes to the part of users of robotic solutions but also to the part of their suppliers, i.e., manufacturers of mobile robots. Users increasingly prefer more complex mobile robotics solutions, with autonomous intelligent control, localisation and navigation.

The behaviour of future collective robots must resemble the behaviour of living organisms. From that point of view, we have to distinguish the concepts of robotics swarm and collective robotics.

Swarm robotics include a set of relatively simple, homogeneous robots. The behaviour of such robots imitates the behaviour of simple living organisms (we refer to them as swarms or flocks) such as bees, ants, or flying birds. For the collective behaviour of such a swarm, relatively simple rules apply. Each member of the swarm has a specified range of activities, which it carries out in favour of the whole swarm [14].

Collective robotics usually involves many, often very heterogeneous robots. Heterogeneous robots may include a whole set of autonomous robots, not requiring a human operation, from mobile robots, road robots, through to flying (drones) and floating robots. These robots possess strong autonomous functions, intelligence, and mobility capabilities. Such robots work with intelligent sensory networks and computer systems organised into cloud-based solutions. In the complex management of collective robots, it is no longer possible to use classical, centralised management. The results of the research in progress have shown that the management of collective robots will require a "proprietary" operating system [15].

The cooperation of collective robots differs significantly from the cooperation of simple swarm robots. In performing complex tasks, in a challenging and unfamiliar environment, collective robots must use distributed control mechanisms that can combine the behaviour of individual, autonomous robots into the complex behaviour of the entire group of robots. We refer to this behaviour as "holonic". For the control of the holonic systems, it is typical to use agent access and multi-agent systems (MAS). The process of cooperation of the group of individual and autonomous robots creates a higher level of collective intelligence, which we call emergence.

In the human body, we can change all the organs except the brain. Its change (disintegration structures and remastered) is blocked. Likewise, the company. Most structures change when reconfigured, but the central control system remains unchanged. It is possible, like the brain, only to expand its function (augmentation) through external expansion. Its architecture must be designed to reflect future changes. The custom control architecture remains to be maintained when reconfigured. A reconfigurable enterprise tries to behave like a living organism [16]. New manufacturing concepts are developed as a response to this paradigm.

*2.3. New Manufacturing Concepts Designed for Factories of the Future*

All new manufacturing concepts seek to meet one of the main objectives and, thus, adaptability, the ability to react immediately to rapid changes in the environment, is also referred to as turbulence. Adaptive manufacturing systems are, at present, a peak of scientists' efforts to formulate the contours of the future production environment. In order to meet the requirement of adaptability, it is possible to approach this in several ways, so scientists have developed and tested a whole group of new manufacturing concepts such as:

- Reconfigurable manufacturing systems
- Competence islands
- Multi-agent control systems

The manufactured product will behave in new manufacturing concepts as a smart entity, able to communicate with its surroundings and able to organise its processing entirely autonomously. Such a product will itself determine the sequence of its processing, allocate the required capacity in the relevant competence islands and sump a mobile robot to ensure its transport in production. To enable such a system of organisation to work safely and reliably and to fulfil the required tasks, it will require new ways of manufacturing planning and control. Next, the seemingly "chaotic" world of production will no longer operate current push control systems. With a vast number of smart elements (entities) in the manufacturing system, there will be complicated relationships and situations that are no longer able to deal effectively with today's hierarchical management. Complex relationships between individual

entities cause a status called emergence, that is, the state in which it will no longer be challenging to predict the future behaviour of such complex systems. Therefore, researchers are experimenting with new management approaches based on the relative autonomy of the individual elements of the manufacturing system and their behaviour, which will resemble the behaviour of intelligent, living organisms. In production, in addition to real objects, there will also be their virtual representatives, which we now refer to as digital twins. Such a dual representation of production is also known as virtual manufacturing.

### 2.3.1. Reconfigurable Manufacturing Systems

The Reconfigurable Manufacturing System (RMS) is a production system, the structure of which is merely adjustable, with the possibility of scaling capacity and flexibility bounded by the selected product family [11]. Figure 3 illustrates the vision of reconfigurable manufacturing systems.

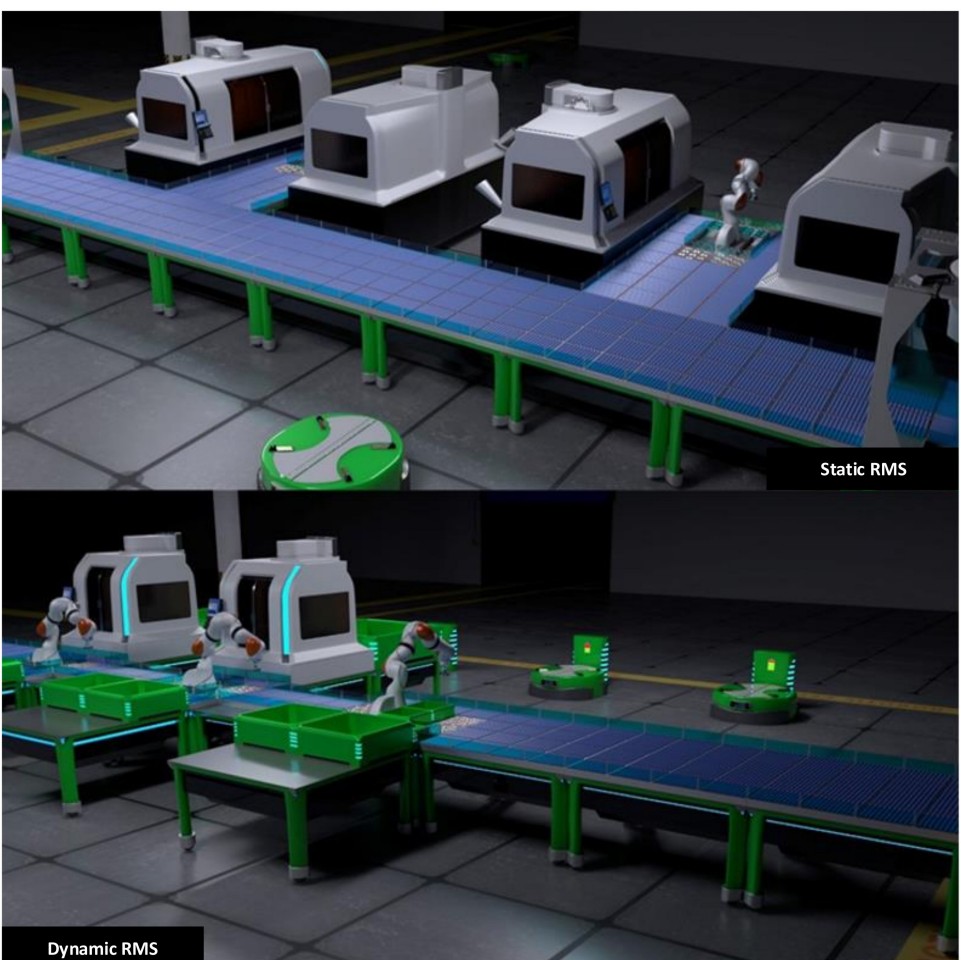

**Figure 3.** Comparison of the static reconfigurable manufacturing system and the dynamic reconfigurable manufacturing system [12].

Reconfigurable manufacturing systems represent the evolutionary phase of the development of manufacturing systems. Their application requires a new approach in which they play a dominant role in reconfigurable machines, jigs, tools, logistics, and reconfigurable control systems [11].

RMS is built to allow for easy and rapid conversion (reconfiguration). This feature pushes reconfigurable manufacturing systems into the adaptive systems area. Reconstructions enable the production system to be adapted to new product types (functionality) and new production quantities (capacity) [17]. Reconfigurability has thus become a new technology that can better meet market fluctuations and turbulence through the gradual rebuilding of the manufacturing system.

Reconfigurability represents the operational ability of the manufacturing system to adapt its functions and capacities to a particular product family.

It results in the desired flexibility of the manufacturing system. As opposed to reconfigurability in the manufacturing system, flexibility is firmly defined. Reconfigurability and elasticity make the adaptive ability of the manufacturing system, which is achieved through a change in its structure. Such a structural change makes it possible to adapt the functions and capacity of the manufacturing system to new requirements. The condition for effective reconfigurability is the requirement to minimise the effort undertaken and maximise the reduction in the time required for the implementation of the changes [11].

### 2.3.2. Competence Islands

The existing large-scale production method, organised rhythmically in production halls and working in the production cycle time, will no longer be able to respond to future customer requirements. Today's "static" production and assembly lines will be replaced by a set of autonomous workplaces called competence islands (Figure 4). It can imagine as virtual production lines, formed dynamically and virtually based on real needs. The competence islands will be equipped with technologies and cooperative robots capable of working safely and reliably with people [18]. Figure 4 illustrates the vision of the competence islands.

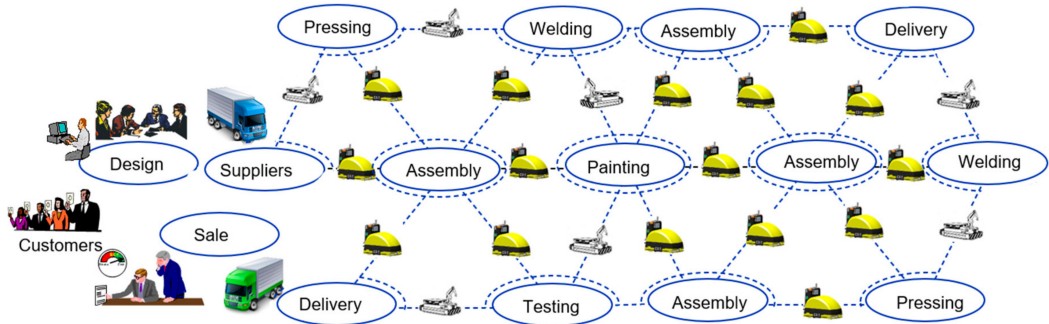

**Figure 4.** Concept of competence islands.

New manufacturing systems should, therefore, be designed as small, highly flexible production units, which will be deployed where there is sufficient real demand. Such manufacturing systems will be designed for the production of the selected product family, which requires that their concept be built on the principles of reconfigurable manufacturing systems.

The activities of future manufacturing systems will be organised differently. Classic production and assembly lines will only be maintained where it is still economically advantageous. Future production will seem to be complete chaos to the outside observer. It will seem that materials, intermediate and elaborate production, and mobile robots are moving unplanned and chaotically. However, each of them will be guided by a strict logic of the parent level, which will enable it to be relatively autonomous. It will, therefore, be organised chaos. For production management, the principles observed from nature, which offer an evolution of proven, optimal practices, will be used.

Holding a strong position in future factories will be intelligent mobile robots and mobile robotic systems and platforms. Thousands of such robots will ensure the movement of the worked products and their processing in a seemingly chaotic world.

The production will be organised as a living organism resembling an anthill, in which the ants appear messy but are strictly organized and specialized, and each of them performs precisely defined tasks that ensure the survival of the anthill.

The product, production equipment, technology and the entire production system will be changed. Manufactured products, manufacturing equipment and mobile logistics means will become intelligent

and communicate with each other. In real-time, they will exchange and share all the necessary data and information.

Mobile robots, transporting a staged product, will move between the competence islands, while the product itself will determine the required operations and plan their order. The observer will not see the classic production line; what will be observed is the apparent physical chaos. However, there will be a hidden virtual line (its digital and virtual data model) made up of the competency islands required for the production of the customer product.

Future production will not be structured according to the production rhythm line, as is the case today, but according to the content of the work to be done. Functional relationships and not fixed cycle times will have a decisive role. This type of production environment will be suitable not only for small production companies but will be particularly advantageous for those types of products that work with high volumes, highly variants of production and which aim at high flexibility and efficiency. Such systems will be able to react more effectively to fluctuations in demand and rapid changes in the models produced by requiring different production technologies. The company Audi claims that the production islands will be much more efficient than today's linear concept.

### 2.3.3. Virtual Manufacturing and Intelligent Agents

The simulation model, detailed, hierarchical and more leveled, containing all the significant factors of the production process, will allow a new type of management, which will be built on dynamic analysis and prediction. If we link such a model to the information sources of production and its sensory system, it will operate as a human organism and will behave adaptively while using real-time data. It will work with its own "physical map", similar to the human body. Today's experiments with in-memory computing are about such future management systems.

In the area of virtual manufacturing and intelligent agents, we meet with the solution of syntactic, semantic and pragmatic boundaries [19]. The first aspect is syntax, which is important for the machine to machine communication. The communication capabilities of agents in multi-agent systems (MAS) are characterised by data exchange mechanisms based on proprietary messages in the form of Extensible Markup Language (XML) syntax and according to MAS standard communication models, for example, defined by the Foundation for Intelligent Physical Agents (FIPA). For establishing CPS in manufacturing environments, the usage of web services is inevitable for the realisation of scalable information exchange. Thus, in addition to a language that describes the information and provides data syntax and semantics, a common underlying mechanism for transfering the information from one entity to another or to perform interactions is needed [20], so the second aspect is semantics. In semantics, we find ontology, annotations, and definitions. The semantics give a mathematical meaning to formulas that, in theory, could be used to establish the truth of a logical formula by expanding all semantic definitions [21]. To provide a proper description of an agent that is readable, understandable and interpretable by other agents in an integrative manner, the description model of each agent needs to follow common design principles, e.g., by making use of a common ontology description, fixed namespaces for agent capabilities (among others) is needed [22]. According to [23], communication between agents can be realised if all agents can find and identify each other and all agents make use of a message system with a predefined ontology, which every agent can understand. In [22], one desired goal to deal with high amounts of raw data from the shop floor would be an automatic assignment of information from the lower levels of the factory. Automated annotation of production information with context information, such as metadata, would reach both machine-readable and interpretable information for autonomous process optimisation as well as a data basis understandable by humans. The third aspect is pragmatics, which means the question of how to use axiomatics to justify the syntactic renditions of the semantical concepts of interest. That is, how best to go about conducting a proof to justify the truth of a CPS conjecture [21]. That means that we must define how to use axiomatics to justify the truth.

New sensory systems allow for the end of the transition from static monitoring systems (sampling and data collection at an interval of one day) to dynamic monitoring (sampling in microseconds, as it

is done today in the process industry). The average values of output parameters (statistics) must be replaced in the new generation monitoring systems with the immediate values and trends of changes in the last, most significant periods.

The monitoring system must include a watchdog, a function that will trace (seek) potential problems, an early warning system that notifies the occurrence of potential problems and an automatic correction mechanism that resolves the potential problem before its real emergence.

If we have enough data about the production system, we can, thanks to virtual reality, create a virtual image of production (its dynamic hologram) and then in such a "reality", virtually track the effect-change factors (visualise them), observe future status and decide on the changes that will be made. In long-enough time, such a system can gradually learn, with the support of a machine-learning system and knowledge system, how to adapt to changing surroundings. If a person makes a decision instead of using computers, manual management is applied. In direct management, in automatic mode, the direct control system decides, and manual interventions are replaced by automatic steering. In the case of manufacturing control, the virtual twin of each real object will be represented by an agent. We refer to a large group of such agents and their management as multi-agent systems (MAS) [24]. Future production will be represented by two worlds: the real world's and its virtual reflection, also called the virtual world. These worlds will be mutually integrated through data. Production data will be collected and processed in real-time. Almost immediately, information about each object in the production will be available—what it is doing, in what state is it located, what is further planned, and what is lacking. The status of each product, machine, tool, device, jig, robot, or person will be immediately scanned, and the processed information will be sent to the control centre. This information will be compared with the next step in the processing of the products, the sequence of future steps will be generated and the system will make the necessary decisions for further processing of the product. The virtual world will allow, if necessary, the simulation of future status and prediction of the effects of the necessary control actions.

## 2.4. Routing of Modelling and Simulation

The basic principle of simulation lies in the simplified representation of the real system of its simulation model, describing only those characteristics of the real system that interest us in terms of its study (simulation). Instead, it would be possible to say that a simulation is a supportive tool that allows the experiment to test the effects of its decisions on the simulation model. By this, we can obtain an answer to the question "what happens if". The great advantage of this approach is that it is possible to previsualise the future behaviour of the system and to realise the necessary interventions in the real system based on its knowledge [25].

In view of the future needs of the simulation, supporting strategic decision-making, new classes of simulation systems must be developed to enable work with aggregated data at different hierarchical levels of the systems being analysed. Such solutions will require the development of entirely new integrated, hierarchical simulation systems capable of modelling complex corporate systems and working with heterogeneous modelling approaches [26]. The main task of the creators of such systems will be to integrate heterogeneous environments into a single, holonic concept. The hierarchy will require integration at micro-, meso- and macrolevels. The simulation environment will provide modelling techniques and approaches for modelling of all corporate hierarchical structures.

Digital twin (DT) is the concept of the functioning of future production systems, based on the digital technology application currently promoted by Siemens. Although the principles of digital twins are known to be more distant, Siemens has stretched the development into a phase of products that are now offered on the market. The digital twin is now presented mainly at the product level, and its essence consists in the creation of a virtual (digital) model of a developed product, machine, or device. The virtual model thus created (digital twin) can be used in all phases of the development, operation, and improvement of the product. For example, the digital twin of a car allows the costs of developing and testing a car to be reduced. The entire development and most of the tests can be implemented

through virtual testing and simulations, using the digital model. Physical tests are used only for the calibration of the test method [27].

The concept of the digital twin has been gradually expanded from product level to process levels, manufacturing systems to the enterprise level. The digital twin can be used in the performance of many business processes, whether it is logistics, manufacturing, assembly, and machining [28]. Industry 4.0 requires phenomenon twins to functionalised the relevant systems (e.g., cyber-physical systems). A phenomenon twin means the computable virtual abstraction of a real phenomenon [29]. The digital factory includes digitisation of the three most important business areas: products, processes and resources. Thus, the era is launched, in which all critical physical production entities are represented by their digital copies and digital models, also called the digital mock-ups (DMUs). In addition to the real manufacturing system, a digital manufacturing system, which is represented by a set of static, kinematic and dynamic digital models, which is integrated into a single digital development environment, i.e., a digital factory, will also be available to all companies.

This has allowed us to study and analyse the efficiency and performance of production before putting it into real operation. Decision-making has begun to become more and more algorithmised, with the database for decision making being the results of dynamic computer simulations. Hence, the beginning of 21st-century enterprises are confronted with two parallel worlds, real and digital (a real-digital world) [27].

Sensor hardening, the rapid development of new communication equipment and systems, have enabled the virtualisation of the world of manufacturing. Such a virtual manufacturing world has generated vast amounts of data that businesses have kept, analysed and started to use for predicting the future behaviour of manufacturing systems. Virtualisation, in this case, means that the managers obtain information about the immediate state of the manufacturing system through sensors. Data from sensors, processed by intelligent algorithms, create a dynamic, virtual image of a real production, which is named "virtual factory", and represent the duality of the real–virtual world (Figure 5). By linking digital, real and virtual worlds, this new quality is now known as the digital twin.

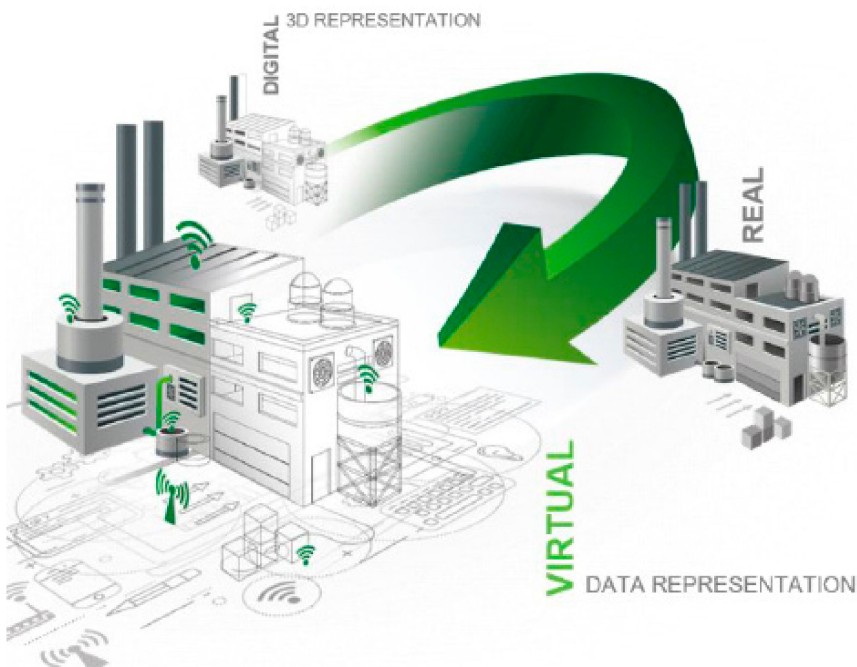

**Figure 5.** Combining three worlds—digital, real and virtual [30].

## 3. Results

### 3.1. Control and Simulation in the Processes of Factories of the Future

The manufacturing system is a multi-factor system. Its model is dynamic, not static. Therefore, it is not possible to say that the efficiency of the manufacturing system is a function of low stock or short, intermediate periods. The efficiency of manufacturing depends on a set of (huge) factors that are dynamically changing over time and are different for each manufacturing system. Although we do not have to know in detail the functioning of each element of manufacturing and we do not have to understand it fully, we can control it. However, we only apply its effectiveness to a very narrow range of criteria (most significant parameters) [30].

Correlation is a statistical characteristic of the statistical dependency rate of two (or more) statistical variables (random quantities). If we consider only two variables, we can easily interpret the dependencies. However, if we move in n-dimensional space with hundreds of variables, relationships will begin between variables (statistical dependencies) to acquire an often meaningless character. In manufacturing systems, we work in reality with an almost infinite number of variables (factors). Therefore, it is very complex (if not impossible) to compile an exhaustive mathematical model of the manufacturing system that would faithfully and accurately represent its dynamism. In this case, the approximate method of computer simulation will help. A cause and its effect, represented by correlation, does not always reveal the causes of the latter, and, rather, may reveal the consequences. Too much data brings the so-called "elusive correlation". A lot of data is used for many different estimates and predictions [31].

The control concept that uses virtualisation contains predictive mechanisms that enable the control system to "see potential scenarios for the future" [31]. The data structure for such a control concept is illustrated in Figure 6.

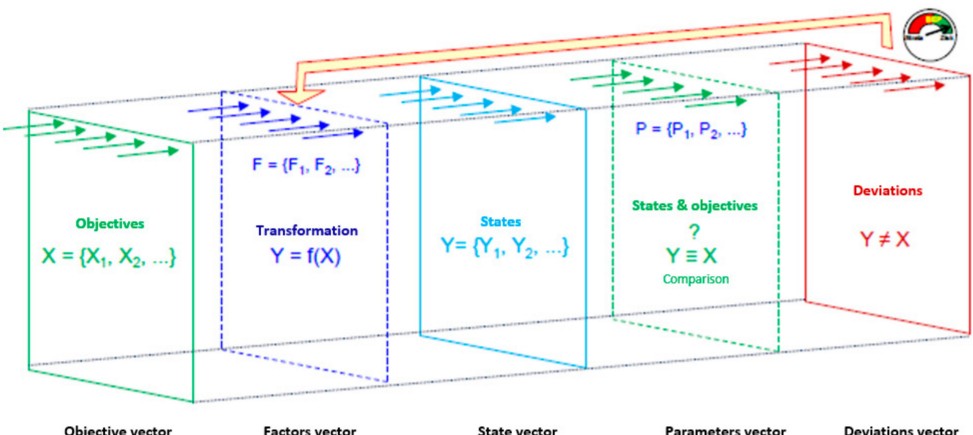

**Figure 6.** The data structure of the factory control system.

The use of a multi-agent control system is for the distribution of tasks and the hierarchical behaviour of the members of the system. In such a system, the holonic system works; it can be seen as a system consisting of subsystems, but at the same time, the system is part of a larger whole (system). A set of holons with their characteristics creates a holonic organisation called holarchy, which is characterised by the fulfillment of common objectives. Holarchy allows the creation of structures and representations of the behaviour of complex systems, often referred to as social systems. The functioning of the holonic systems is based on the use of the ability of autonomous agents. An agent is a system entity that has a specific degree of independence, allowing it to autonomously address tasks within a defined level of action. Agents accept tasks from the parent level of the holarchy, but their solution is carried out autonomously.

An intelligent agent is a computational or natural system capable of perceiving its surroundings and, based on its monitoring, performing actions that result in the extreme of its objective function (minimum, maximum), thereby fulfilling the global objectives of the system. In the agent systems, in the vast amount of interactions that occur between individual, autonomous agents (for example, in social systems), we are no longer able to predict the future behaviour of such a system [32]. If we were modelling such a system, it would be better to define the behaviour of individual parts of the system (agents). An agent can use the services of holon, which is used for simulation of varying inputs and to see the outcomes of actions. The use of simulation metamodelling within the holon simulation is illustrated in Figure 7.

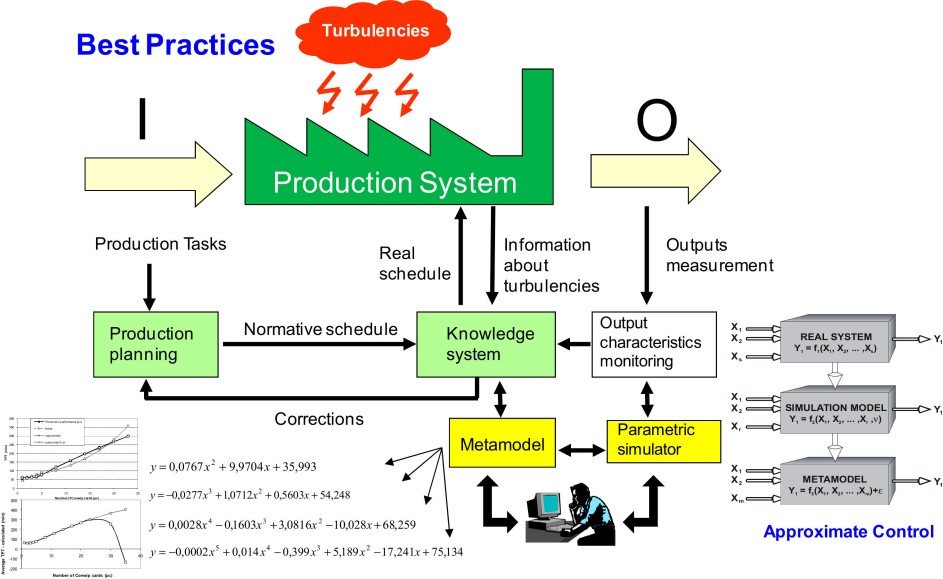

**Figure 7.** Use of simulation metamodelling in the manufacturing control of complex manufacturing systems.

Figure 8 also illustrates the principle of application of digital factory instruments to changes in the product range, the exchange of technology and the change of layout. As seen, the entire management concept is first developed and tested offline in the virtual environment of the digital factory. Agents that represent the physical elements of the system use the knowledge of previous actions as well as existing models and, on the virtual model of the manufacturing system, carry out experiments in which scenarios are tested. Then, it selects the appropriate scenario that matches the target characteristics of the system. After completion of the development, the validated control concept is transferred to the real production system. Therefore, the simulation becomes an emulation when the startup point of a real-element agent that is recorded in a specific position predicts future statuses. The principle of the knowledge-based environment will support the system in the form of learning from process activities.

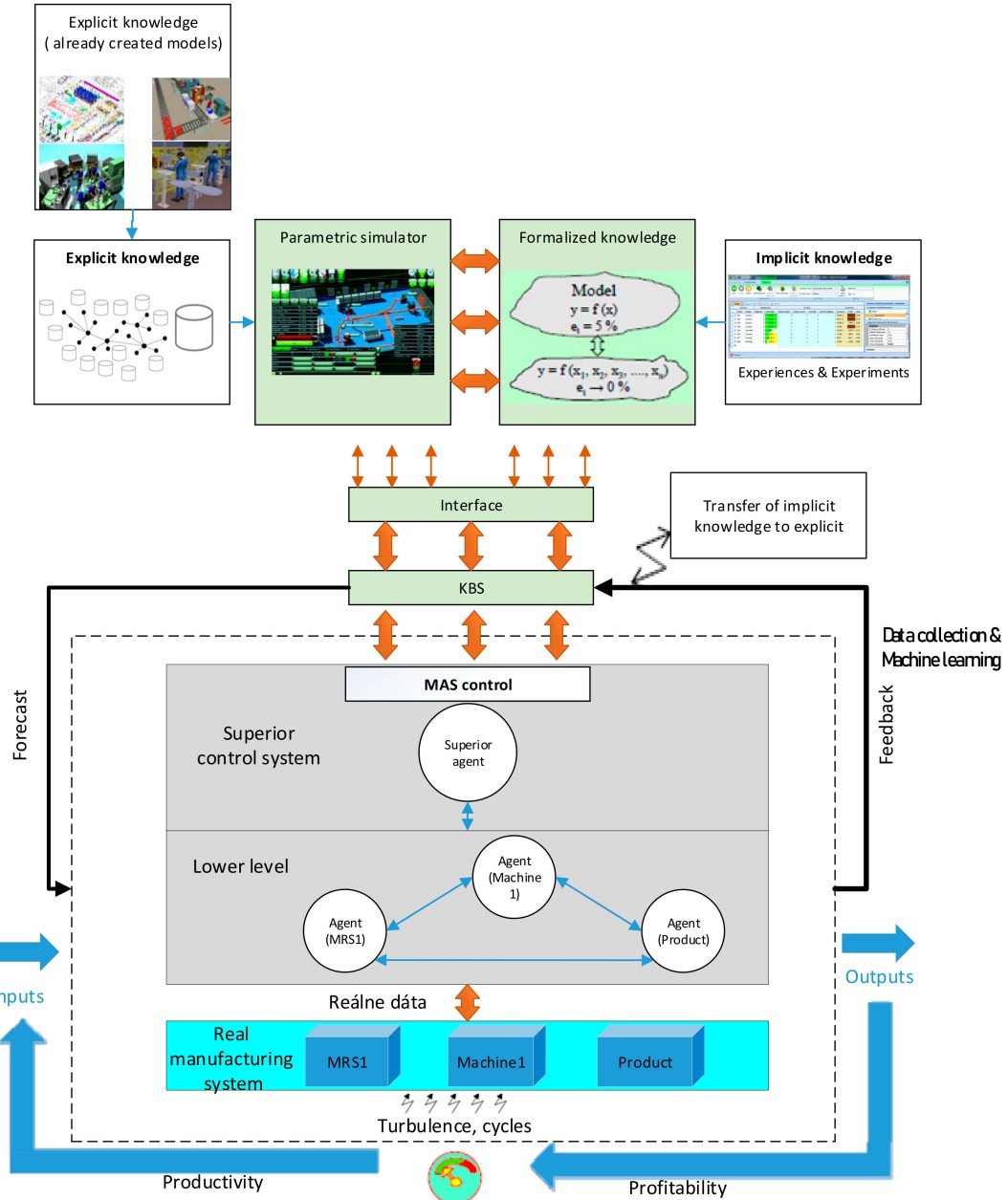

**Figure 8.** Linking simulations in continuity to control and learning from processes.

### 3.2. Application to the Zilina Intelligent Manufacturing System

The question involved was applied to the Zilina Intelligent Manufacturing System (ZIMS), whose structure is illustrated in Figure 9. The Zilina Intelligent Manufacturing System (ZIMS) has been built to faithfully represent advanced manufacturing systems with its practical design, while also enabling experimentation and further research in the field of intelligent manufacturing systems. The whole concept of ZIMS was designed as a holistic system. Individual holons represent the main subsystems and elements of advanced business systems. As part of the simulation application design, a control system based on simulations, emulations and metamodelling was applied.

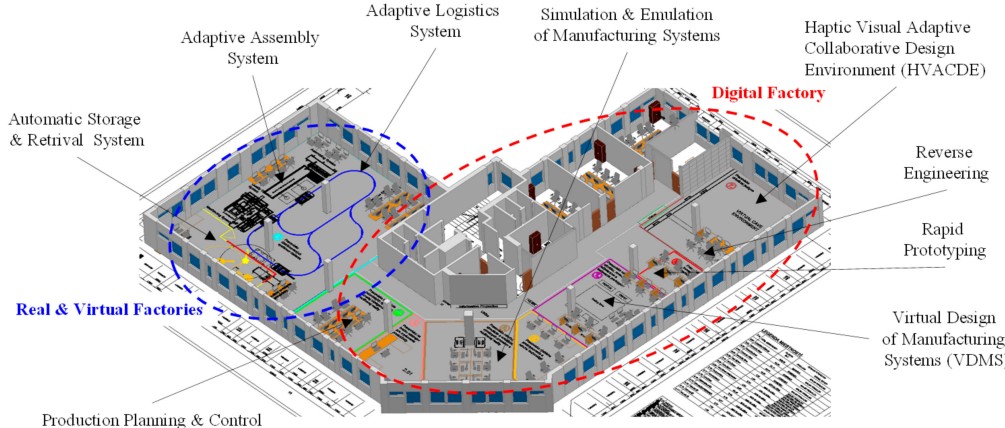

**Figure 9.** Structure of the Zilina Intelligent Manufacturing System (ZIMS) laboratory.

The logic of a more detailed subdivision of the holons is shown in Figure 10, in which the modelling and simulation are under holon manufacturing.

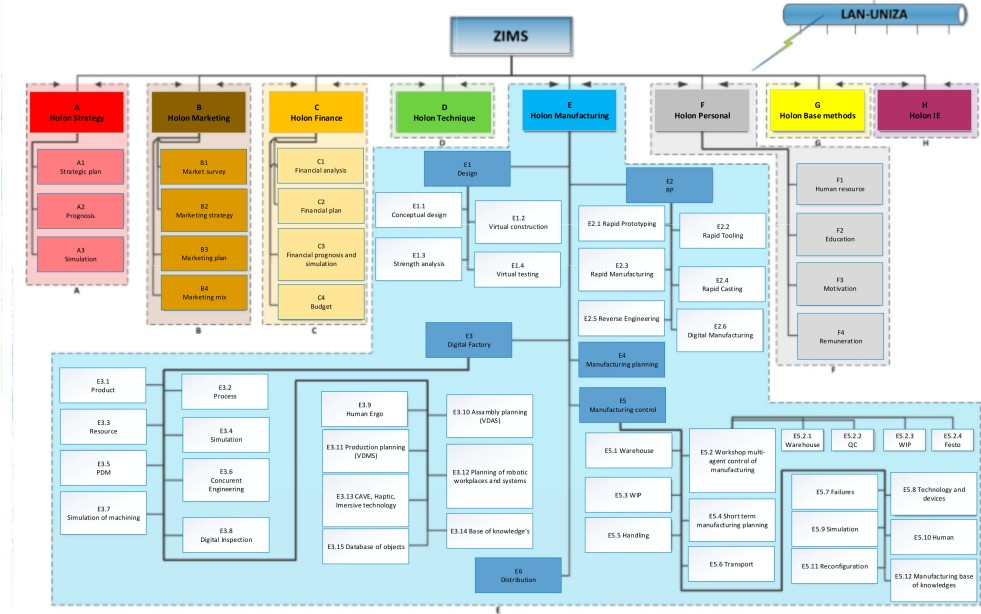

**Figure 10.** A more detailed breakdown of the holons of the ZIMS laboratory.

As seen from Figure 11, the structure of the holonic control respects the functional requirements of the enterprise control system. Different data connections and standards are used at the level of the individual holons: STEP, IGES, PNG, RAW, RGB, VRML, DXF.

Communication within the holons is illustrated by the example of manufacturing (Figure 12).

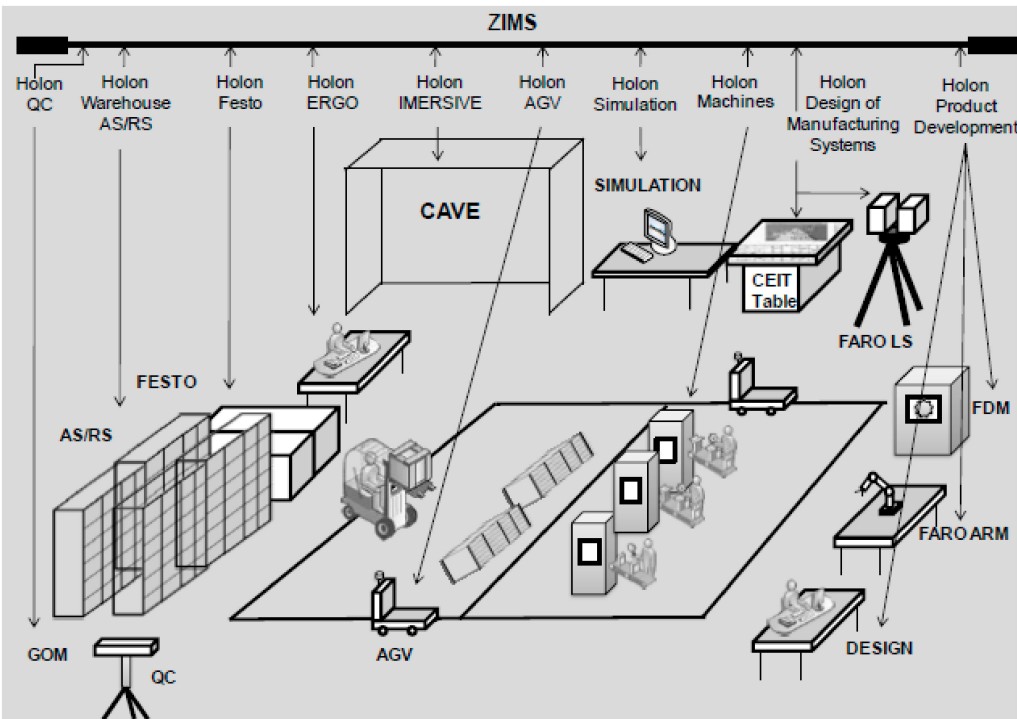

**Figure 11.** Structure of the holon manufacturing in ZIMS.

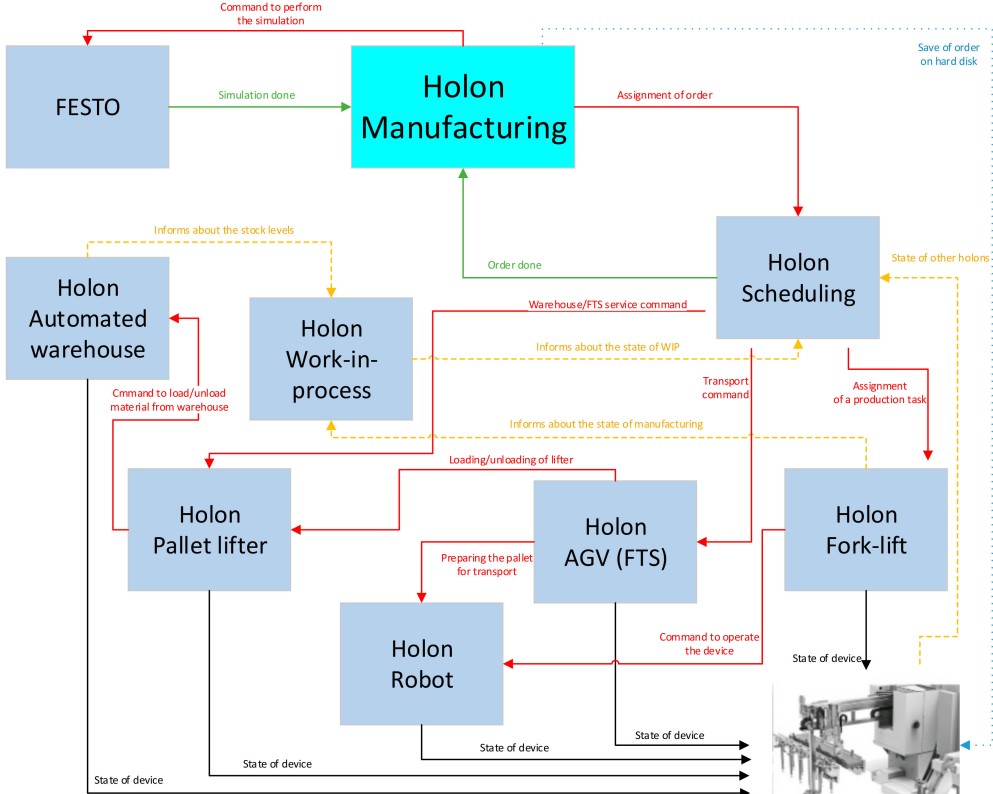

**Figure 12.** Communication of the holon manufacturing control in ZIMS.

### 3.3. Use of Metamodelling in Laboratory ZIMS

In the laboratory, ZIMS is used for the determination of individual holon action metamodelling. An essential part of using the simulation as support for control is to train the simulation network with

data. For the verification of metamodeling, a model of manufacturing cell of the ZIMS concept is created, which consists of three machines. They gradually work on intermediate A, which enters the system at regular intervals, every 3 minutes. It is then transported by a conveyor belt to the buffer of the first machine, S1, from which it is taken and subsequently worked on if the machine is free. When the operation is completed, the semi-finished goods are transported again using the conveyor belt into the buffer of the next machine, S2, and the procedure is repeated. The transport between the last machine, S3, and the "output" is not considered.

The operational times of the machinery and the times of transport between the workplaces (Table 1) are the same at a purely theoretical level; in the event of a failure to run the system, the machines work at 100 (which is unrealistic, but it is only an explanation of the process of working in the formation of the metamodel). However, overall system productivity is affected by the failure of the second machine, S2, which occurs at particular time intervals $X1 = \{x11,x12,...,x17\} = \{15,20,25,30,40,50,60\}$ and the repair time is defined by a set of $X2 = \{x21,x22,...,x26\} = \{5,8,10,12,15,20\}$.

**Table 1.** Times, transport times and the interval of arrivals of intermediate products into the system.

| Machine | Operation Time (min) | Input | Interval of Arrivals into System (min) | Ways | Transport Time (min) |
|---|---|---|---|---|---|
| S1 | 3 | | | c1 | 1 |
| S2 | 3 | A | 3 | c2 | 1 |
| S3 | 3 | | | c3 | 1 |

The denotation of variables is X1—time between failures; X2—repair time; Y—lead time of production.

Then we selected (based on short pilot runs) time simulation, namely, one working week with a single-shift 7.5-hour operation (i.e., 2250 min) and a time of production of 50 min. After completing all these steps, we could proceed to the implementation of simulation experiments.

These input data were performed for all combinations of the levels of factors X1 and X2 mentioned above, which totals 42 simulation runs (Table 2).

**Table 2.** Results of simulation experiments.

| X1 | X2 | Y1 | X1 | X2 | Y1 |
|---|---|---|---|---|---|
| | 5 | 294.17 | | 12 | 333.62 |
| | 8 | 403.55 | 30 | 15 | 386.73 |
| | 10 | 461.68 | | 20 | 461.09 |
| 15 | 12 | 511.68 | | 5 | 137.65 |
| | 15 | 574.15 | | 8 | 200.25 |
| | 20 | 654.48 | | 10 | 237.61 |
| | 5 | 237.62 | 40 | 12 | 271.98 |
| | 8 | 334.37 | | 15 | 318.84 |
| | 10 | 388.11 | | 20 | 386.54 |
| 20 | 12 | 434.56 | | 5 | 114.79 |
| | 15 | 494.27 | | 8 | 167.78 |
| | 20 | 573.89 | | 10 | 199.99 |
| | 5 | 200.62 | 50 | 12 | 230.04 |
| | 8 | 286.01 | | 15 | 271.60 |
| | 10 | 334.50 | | 20 | 333.50 |
| 25 | 12 | 377.58 | | 5 | 98.88 |
| | 15 | 434.15 | | 8 | 144.76 |
| | 20 | 511.66 | | 10 | 173.01 |
| | 5 | 173.45 | 60 | 12 | 199.86 |
| 30 | 8 | 249.62 | | 15 | 237.46 |
| | 10 | 293.80 | | 20 | 293.87 |

Subsequently, after verifying the data from the simulation, we determined the ones that will serve to train the network and those that will be test data. Artificial neural networks (ANNs) were also tested during the learning process, and validation was not necessary [33]. For training, we selected a set of 35 combinations of data obtained from simulation runs Table 3. The training set Table 4 modified the scales, and a generating error was detected using the test set. The entire process of creation, training, testing and validation took place in the Matlab program environment [34].

**Table 3.** Training data (inputs and outputs) for the artificial neural network (ANN).

| X1 | X2 | Y1 | X1 | X2 | Y1 |
|----|----|------|----|----|--------|
| 15 | 5  | 294.17 | 30 | 15 | 386.73 |
| 15 | 10 | 461.68 | 30 | 20 | 461.09 |
| 15 | 12 | 511.68 | 40 | 5  | 137.65 |
| 15 | 15 | 574.15 | 40 | 8  | 200.25 |
| 15 | 20 | 654.48 | 40 | 10 | 237.61 |
| 20 | 5  | 237.62 | 40 | 15 | 318.84 |
| 20 | 8  | 334.37 | 40 | 20 | 386.54 |
| 20 | 12 | 434.56 | 50 | 5  | 114.79 |
| 20 | 15 | 494.27 | 50 | 8  | 167.78 |
| 20 | 20 | 573.89 | 50 | 10 | 199.99 |
| 25 | 5  | 200.62 | 50 | 12 | 230.04 |
| 25 | 8  | 286.01 | 50 | 20 | 333.50 |
| 25 | 10 | 334.50 | 60 | 5  | 98.88  |
| 25 | 12 | 377.58 | 60 | 8  | 144.76 |
| 25 | 15 | 434.15 | 60 | 10 | 173.01 |
| 30 | 8  | 249.62 | 60 | 15 | 237.46 |
| 30 | 10 | 293.80 | 60 | 20 | 293.87 |
| 30 | 12 | 333.62 |    |    |        |

**Table 4.** Test data for the ANN.

| XT1 | XT2 | YT |
|-----|-----|--------|
| 15  | 8   | 403.55 |
| 20  | 10  | 388.11 |
| 25  | 20  | 511.66 |
| 30  | 5   | 173.45 |
| 40  | 12  | 271.98 |
| 50  | 15  | 271.60 |
| 60  | 12  | 199.86 |

After we enter all the input factors and commands to display an error between the outputs of the ANN and the specified Y results, rendering the ANN output differences for the test data and the actual output of YT, a network training order with the training data is entered. When starting the learning process, a Figure 13 window appears, which can be followed by the training process, the number of running eras, the duration of learning, and a shrinking/increasing error [35].

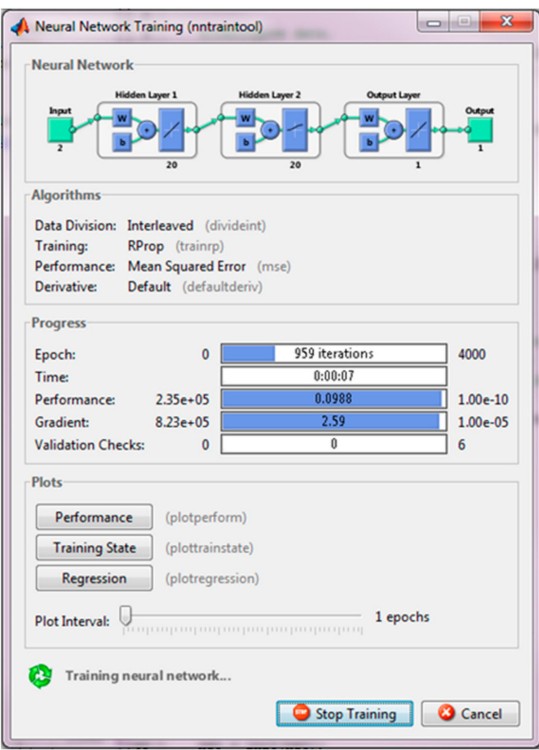

**Figure 13.** Example of the ANN training process.

Once the ANN has been trained and reaches a satisfactory result, it is possible to use the ANN to address specific problems by putting new values into ANN for which no simulation runs have been made, but the responses to them are of interest to us. The validation itself does not need to be carried out since test data have already been used. However, for a demonstration, in Table 5, we compare outputs generated by our ANN for specified input data with simulation outputs.

**Table 5.** Comparison of the results of the simulation and trained ANN.

| X1 | X2 | Y1-Simulation | Y1-ANN |
|----|----|---------------|--------|
| 17 | 5 | 268.51 | 268.6991 |
| 18 | 9 | 387.27 | 387.7591 |
| 22 | 4 | 185.78 | 196.6634 |
| 27 | 11 | 338.75 | 338.5108 |
| 29 | 14 | 379.01 | 378.5756 |
| 35 | 10 | 262.80 | 262.2215 |
| 38 | 16 | 345.35 | 344.4772 |
| 42 | 6 | 153.04 | 154.3730 |
| 46 | 18 | 328.02 | 327.5044 |
| 51 | 13 | 240.94 | 241.3330 |
| 58 | 17 | 267.40 | 266.8995 |
| 63 | 7 | 124.98 | 125.6077 |

## 4. Discussion

Based on the knowledge learned from the long-term research in the field and the practical experience gained in dealing with the projects in the industry, we can anticipate the development of simulation environment requirements for the factories of the future.

Due to the fast onset of solutions of Industry 4.0 and the extensive use of sensors the main task for future simulation environments is the ability to model and simulate the behaviour of complex systems. When using a large number of sensors, processing data in real-time and the autonomous

behaviour of the elements of the manufacturing system, the factories of the future will experience emergent phenomena.

This change will cause the simulation systems today to be used to simulate an emerging complexity. Therefore, one of the crucial tasks for the creators of simulation systems will be to develop solutions to simulate complexity in manufacturing systems. For the dynamism of the autonomous behaviour of the elements of manufacturing systems, the principles of multi-agent systems can be used in simulations, which today represents the agent simulation. Another suggested development will be the effort to "simplify" complex problems, in which way, one of the routes can be the use of simulation metamodelling. Several types of research work addressed in our department declare this development trend.

One of the crucial requirements for a new simulation environment will be its ability to offer the functionality of the emulatory environment. The integration of the real manufacturing system with its digital and virtual models will enable both offline and online optimisation, and the simulation will become part of real-time control systems.

The future simulation environment will naturally reflect the requirements of the factory of the future. In its creation, all modelling and statistical support tools, which are now commonly used in the simulation, will be used.

However, this will fundamentally change the way the simulation is implemented. Three main approaches to simulations (event orientations, process orientation and activity orientation) have traditionally been used, while new simulation algorithms will be built on distributed, autonomous principles. Due to the requirement for the reconfigurability of manufacturing systems, new simulation systems will have to offer entirely new functionalities and thematic templates, as shown in the example of the research and development of the agent simulator for future hospitals or the development of a multi-agent control (and simulation) system of complex logistics systems.

Methods and tools supporting the transformation of physical systems into virtual ones are evolving. The dynamics of the development of such systems are displayed in Figure 14.

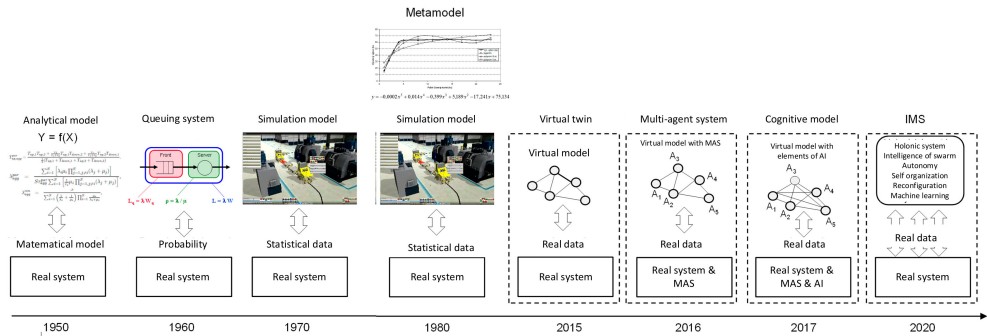

**Figure 14.** Evolution of real system to virtual model [36].

As to further perspectives for the development of modelling and simulation in factories of the future, the Department of Industrial Engineering sees, in the following period, further integration of the various industrial engineering methods and tools used in the industry into computer simulation software tools. The concept of planning and control of future factories, through the use of computer simulations, forecasting of demand and monitoring of enterprise performance indicators (e.g., productivity), must also be developed within the framework of the approach mentioned above. These themes enter into a new dimension because the coordination of the production chain is becoming a prevalent task for all stakeholders, and the aim is to achieve a common synergy effect.

The research conducted has clearly shown that the theory of complex systems, on the basis of the requirements of factories of the future, has progressed considerably and is already providing practical tools for designers of future manufacturing systems. Key contributions of the research base on case applications are:

- Growing performance
- Thanks to metamodelling applications, a quick prediction of emergent properties of the analysed system is achieved
- Identification of bottlenecks
- Identification of obsolete and nonturnover stocks
- Reduction of stocks of finished products
- Increasing key indicators of the enterprise (such as performance, productivity)

In the frame of research limitation, it can be said that we mainly focused on an application on the reconfigurable manufacturing system and the processes within (e.g., manufacturing, logistics). However, the model can be used on conventional manufacturing systems that have a certain level of communication ability and self-awareness.

We focused our current efforts on researching new approaches to simulating complex systems using agent simulation. In the future, we want to focus our research work mainly on the area of intelligent manufacturing systems, which using reconfigurable manufacturing systems, adaptive logistics and the concept of competency islands.

## 5. Conclusions

Manufacturing systems of the factories of the future will have new features that will enable them to respond quickly and efficiently to frequently changing customer demands. These manufacturing systems will be designed as modular, reconfigurable and intelligent holonic systems, capable of rapidly changing their functions and capacities based on auto diagnostics. Designing and analysing the behaviour of future manufacturing systems will require heterogeneous simulation models and new simulation tools, allowing for rapid, comprehensive analysis and interpretation of the results obtained. The future simulation environment will naturally reflect the requirements of the future factory. In its creation, all modelling and statistical support tools, which are now commonly used in the simulation, will be used. However, this will fundamentally change the way the simulation is implemented. Three main approaches to simulations (event orientations, process orientation and activity orientation) have traditionally been used, while new simulation algorithms will be built on distributed, autonomous principles. In view of the requirement for reconfigurability of manufacturing systems, new simulation systems will have to offer entirely new functionalities and thematic templates. By engaging the simulation, it is possible to at least partially estimate the results of the interactions at emergence and to control the manufacturing system. The purpose of the article is to outline the use of modelling and simulation in control of processes in a factory of the future. An application of metamodelling inside a manufacturing holon in laboratory ZIMS was presented as an example. The article in the periphery also describes the developed manufacturing concepts that will be used in the factories of the future, which will meet the demands of the paradigm of mass customisation and personalisation.

**Author Contributions:** All authors contributed to writing the paper, documented the literature review, analysed the data and wrote the paper. All authors were involved in the finalisation of the submitted manuscript. All authors have read and agreed to the published version of the manuscript.

**Funding:** Slovak Research and Development Agency under contract no. APVV-18-0522.

**Acknowledgments:** This work was supported by the Slovak Research and Development Agency under contract no. APVV-18-0522.

**Conflicts of Interest:** The authors declare no conflict of interest.

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
