# Peer review of "Modeling and Simulation of Processes in a Factory of the Future"

_applsci, doi:10.3390/app10134503_

Round 1

Reviewer 1 Report

The authors described how to apply modelling and simulation in the processes of factories of the future. They also described results from an application simulation in the control of processes of manufacturing systems. It is an interesting paper. However, some issues need to be provided.

In section 3, the Zilina Intelligent Manufacturing System (ZIMS) has been built to faithfully represent advanced manufacturing systems with its practical design, while also enabling experimentation and further research in the field of intelligent 594 manufacturing systems. More quantitative results from this system should be provided and discussed.

The reference format is not consistent.

Author Response

Dear reviewer,

Thank you for your valuable comments. Base on them we add part of the result about the application of metamodeling. We also check and complete references.

Kind regards

Collective of authors.

Reviewer 2 Report

The factory concept and description of knowledge that the authors presented is quite misleading given the fact of current literature of Industry 4.0. The main problem of Industry 4.0 is to deal with three aspects simultaneously. These three aspects are syntax (machine to machine communication, or simply automation), semantics (what is what, ontology, annotation, etc.), and pragmatics (problems open system relevant issues such as trustworthiness). At the same time, knowledge classification, signal processing, etc. are major issues that make the cyber-physical systems functional. Digital twining of objects, processes, and phenomena are also important aspects.

Author may refer to the following articles and rewrite some parts of the manuscript to make it more presentation of the current context of manufacturing system related research.

Modeling and simulation of complex manufacturing phenomena using sensor signals from the perspective of Industry 4.0 (https://doi.org/10.1016/j.aei.2018.11.003)

Fundamental Issues of Concept Mapping Relevant to Discipline-Based Education: A Perspective of Manufacturing Engineering (https://doi.org/10.3390/educsci9030228)

Machining Phenomenon Twin Construction for Industry 4.0: A Case of Surface Roughness (https://doi.org/10.3390/jmmp4010011)

Hidden Markov model-based digital twin construction for futuristic manufacturing systems (https://doi.org/10.1017/S089006041900012X)

Author Response

Dear reviewer,

Thank you for your valuable comments. Base on them we completed part about Industry 4.0 and change or adjust figures in the article.

Kind regards

Collective of authors.

Reviewer 3 Report

This paper seems to be a lecture text or in the best case a review article.

The whole sections 1 and 2 and a large portion of section 3 (for instance the rows 130-178, 198-213, 263-267, 289-308, 329-338, and 393-481, and Figures 4, 5, and 8) are repetitions from lectures for undergraduate students.

The remaining portion of section 3, and sections 4 and 5 are repetitions of what has been reported and therefore known from before (for instance the rows 614-671)). The Conclusions (section 5) repeat what is known since many years ago and do contain the conclusions drawn by the presenting authors (as the result of their own work).

The authors use the "right" section titles but do not seem to follow the description of what and why in the Introduction, a decsription of how (methods, materials, exerimental procedure) in the subsequent sections, Results, Discussion, and Conclusions. The almost 10 pages long section 2 is called Materials and Methods. However, this method section is just a repetition from undergraduate lectures (and a literature survey) and does not describe which specific methods the authors have used to conduct this study/investigation/work (and why).

Suggestions to the authors:

  • Decide whether this should be a review paper (based on literature survey) or an article/a paper describing your recent work.
  • If your decision is to contribute with an article, please revise the current version by considering the above-mentioned comments.

Author Response

Dear reviewer,

Thank you for your valuable comments. Even that may look that this is lectures for undergraduate students it is a new thing from our research and contain long year experience of authors which worked with ZIMS. We try to incorporate your and other reviewer suggestions in maximum scale and hope that it improves the quality of the article.

Kind regards

Collective of authors

Reviewer 4 Report

This paper focuses on the development of the smart factory with the introduction of modelling and simulation approaches. The case applications were conducted. This paper sounds interesting. However, there are some suggestions for improving the quality of the paper as follows:

(a) the abstract should provide more specific in terms of the case application and results obtained.

(b) What are the practical implications and limitations of this research study?

(c) What are the key contributions of the research based on the case applications that were conduced? Are there any differences between exiting application found in literature and the proposed case applications?

Author Response

Dear reviewer,

Thank you for your valuable comments. Base on them we add part of the result about the application of metamodeling. We also adjust the discussion part about the mentioned suggestions.

Kind regards

Collective of authors.

Reviewer 5 Report

This paper reviews the application of simulation in the modeling of the process in factories focusing on the future trend. The ideas are quite explicitly describe

Comments:

  • 2nd page 2nd para is unclear
  • Page 5, the last paragraph, tells about conventional manufacturing, explains what conventional manufacturing is?
  • 1 does not seem to have a connection with the heading 2; the separate heading might be appropriate for this part.
  • 2 does not describe a proper connection with simulation
  • The scope of the simulation options need to be discussed
  • A figure describing the year wise improvement in the simulation will be appropriate
  • Figure 4 describes the effects of product variety, showing other effects of product variety will be an added advantage

Author Response

Dear reviewer,

Thank you for your valuable comments. Base on them we adjust the introduction and methods part of the article base on suggestions with aim on figures.

Kind regards

Collective of authors.

Reviewer 6 Report

The paper is very interesting and meets all the requirements for a good scientific publication. 

More in detail:

The paper contains new and significant information adequate to justify publication.

The paper demonstrates an adequate understanding of the relevant literature in the field and cites an appropriate range of literature sources. No significant work is ignored.

The paper clearly expresses its case, measured against the technical language of the fields and the expected knowledge of the journal's readership. The attention has been paid to the clarity of expression and readability, such as sentence structure, jargon use, acronyms, etc.

The conclusions tie adequately together with the other elements of the paper.

Comments:

Please, add limitations of the study and suggestions for future research.

Author Response

Dear Reviewer,

Thank you for your review report and suggestions. Base on them, we complete manuscript text.

Kind regards,

Collective of authors

Round 2

Reviewer 1 Report

The authors answered my comments. However, there are many grammar errors in the text. This manuscript needs a technical writer to edit it.

The reference format is not consistent. For example:

"Mleczko, J.; Dulina, L. Manufacturing Documentation for the High-Variety Products. In: Management and 923 Production Engineering Review. Warszawa, 2014. 5(3). Pp. 53-61. ISSN 2080-8208, eISSN 2082-1344"

This is a journal paper. It should be

"Mleczko, J.; Dulina, L. Manufacturing documentation for the high-variety products.Management and Production Engineering Review 2014, 5(3), 53-61."

Please edit all references.

Author Response

Dear reviewer

Thank you for your current and previous valuable suggestions. Base on comments we formatted reference in software Zotero in ACS Style and also check text by software Grammarly.

Kind regards

Collective of authors

Reviewer 2 Report

Throughout the paper, we see a mix of American and British spelling. Authors have mentioned job title of some individuals in the manuscript. This is strange.

There is no logical order in the contents. Sometimes, the contents do not make any sense.

For example, consider the abstract. The authors describe in the last two sentences, only, what they want to do in this paper. In this case, the other sentences are redundant.  The same argument is true for all sections in the manuscript.

Authors introduced the notions of explicit knowledge and implicit knowledge (Figure 13). These notions refer to a well-known definition of knowledge in organization science: knowledge can either be of the tacit or explicit types. Tacit knowledge pertains to intuitions, experiences, and know-how possessed by active individuals in their respective organizations. Consequently, it is challenging to identify or even codify such knowledge. Explicit knowledge includes documented instructions for facilitating organizational activities. It is, therefore, easy to identify and share. Tacit knowledge dynamically transforms into explicit knowledge and vice versa through social or teamwork-based interactions (dialogue) among employees. Remarkably, such transformations do not require formal logical processes to be performed. This contradicts the definitions of knowledge reported in other disciplines, such as information science. However, other schools of thought in organization science exist related to knowledge and its formation. For example, some authors consider that there are five types of knowledge—scientific, quantitative, qualitative, tacit, and intuitive.

As such, the reviewer is confused seeing that the authors have defined simulation as implicit knowledge.

Therefore, the reviewer is not in a position to accept the work as an article. Please focus on a specific topic and make your points so that it makes sense.

Author Response

Dear reviewer

Thank you for your valuable comments. Base on comments we revised the manuscript in the following directions

  • Mix of American and British spelling was removed, and text is now in British spelling
  • Base on the suggestion of logical order of contents, we adjust the text to be more in logical sequence chapter Industry 4.0 was divided and moved to other chapters. Some parts that were repeated several times was reduced.
  • Base on the suggestion we completed to text, what was the primary purpose of the article and links between chapters.
  • Figure 13. was adjusted to be more truthful

We hope that you in the revised manuscript will find the suggested changes.

Kind regard

Collective of authors

Reviewer 3 Report

As far as I understand, this paper is not intended to be a review paper. If so is the case, a major revision is required. Currently, the paper can be classified as a literature survey which contains basic descriptions of Industry 4.0, product-process matrix, virtual manufacturing engineering, simulations/modelling, control systems… Both the text and the figures are basic. For instance, Figure 9 (formerly Figure 8) is one of the first figures shown in high school programs.

It is not possible to see what the authors’ idea/concept is, the methods they have used, the results from their investigation etc., as scientific/technological papers are normally outlined.

I would suggest that the authors revise the paper in accordance to the following:

  • Abstract (a brief description of what the authors have done),
  • Introduction (the problem/issue/concept the authors have focused on),
  • Materials and methods (how they solved the problem),
  • Results (what they found out),
  • Discussion (how one can interpret the findings, what the results mean),
  • Acknowledgments, and
  • References.

Please see also my previous comments (the so-called Report 1 comments).

Author Response

Dear reviewer

Thank you for your valuable comments. Base on comments we revised the manuscript in the following directions

  • For all text was applied British spelling
  • We try to revised text chapters according to your suggestion. Base on suggestion if some parts are well-grounded we adjust the text to be more in logical sequence chapter Industry 4.0 was divided and moved to other chapters. Some parts that were repeated several times was reduced. Ad some insubstantial parts were removed where we came from your previous comments.
  • Base on the suggestion we completed to text, what was main purpose of the article and links between chapters and how they fit together.
  • Base on the suggestion, we remove some figures that can be seen as insubstantial.

We hope that you in the revised manuscript will find the suggested changes.

Kind regard

Collective of authors

Reviewer 5 Report

Thanks for the revisions. 

Author Response

Dear reviewer

Thank you for your previous valuable suggestions.

Kind regards

Collective of authors

Round 3

Reviewer 2 Report

The reviewer still see mixing of American and British English. Title and Figures have American English. British English is followed in the main text of the manuscript.

If the title of the manuscript is "Modeling and simulation of processes in factory of the future," the we expect the following

how to model and simulate processes in a factory in general?

what are the techniques to build models of processes?

what are the techniques to simulation systems of processes?

what does the author means by process? is it manufacturing process? is it assembly process?

why we can do simulation using functions, f(x)? how about we do not have numerical data?

this means that the authors need to focus on something concrete. now the canvas is very wide.

Author Response

-

Reviewer 3 Report

I have suggested in 2 reviews that the authors revise the paper in accordance to the following:

- Abstract (a brief description of what the authors have done).

   This description is still missing.

- Introduction (the problem/issue/concept the authors have focused on).

   Only rows 135-145 can be considered as an “Introduction” and rows 140-144 as a “problem/issue/concept description” for

   this paper.

- Materials and methods (how the authors have solved the problem/addressed the issue).

   This section of the paper is a literature review. It is not possible yet to read how the authors have solved the problem/

   addressed the issue…

- Results (what they found out).

  This section begins with a literature review and contains figures that cannot be considered as results related. Figures 7, 8, 9,

  and 15 cannot be considered as results (of this investigation)… (Figure 9 includes a non-English expression.)

- Discussion (how one can interpret the findings, what the results mean).

- Conclusions

   It is not possible yet to read exactly what the conclusions are. Rows 1067-1081 are repetitions of what others have done/is

   known. The authors should focus on rows 1082-1087 (these are still descriptive) and rephrase these rows to provide their

   conclusions.

- Acknowledgments.

  Done

- References.

  Done

Please see also my previous comments.

Author Response

-